# Hypertensive Disorders of Pregnancy and Medication Use in the 2015 Pelotas (Brazil) Birth Cohort Study

**DOI:** 10.3390/ijerph17228541

**Published:** 2020-11-18

**Authors:** Lisiane Freitas Leal, Sonia Marzia Grandi, Vanessa Iribarrem Avena Miranda, Tatiane da Silva Dal Pizzol, Robert William Platt, Mariângela Freitas da Silveira, Andréa Dâmaso Bertoldi

**Affiliations:** 1Department of Epidemiology, Biostatistics, and Occupational Health, McGill University, Montreal, QC H3A 1A2, Canada; sonia.grandi@mail.mcgill.ca (S.M.G.); robert.platt@mcgill.ca (R.W.P.); 2Programa de Pós-graduação em Saúde Coletiva—PPGSCol, Universidade do Extremo Sul Catarinense, Criciúma 88806-000, Brazil; vanessairi@gmail.com; 3Programa de Pós-Graduação em Epidemiologia, Faculdade de Medicina, Universidade Federal do Rio Grande do Sul, Rio Grande do Sul 90610-000, Brazil; tatiane.silva@ufrgs.br; 4Programa de Pós-graduação em Epidemiologia, Faculdade de Medicina, Universidade Federal de Pelotas, Pelotas 96020-220, Brazil; mariangelafreitassilveira@gmail.com (M.F.d.S.); andreadamaso.epi@gmail.com (A.D.B.)

**Keywords:** hypertensive disorders of pregnancy, birth cohort, cardiovascular agents, pregnancy, pre-eclampsia, pharmacoepidemiology

## Abstract

Hypertensive disorders of pregnancy account for approximately 22% of all maternal deaths in Latin America and the Caribbean. Pharmacotherapies play an important role in preventing and reducing the occurrence of adverse outcomes. However, the patterns of medications used for treating women with hypertensive disorders of pregnancy (HDP) living in this country is unclear. A population-based birth cohort study including 4262 women was conducted to describe the pattern of use of cardiovascular agents and acetylsalicylic acid between women with and without HDP in the 2015 Pelotas (Brazil) Birth Cohort. The prevalence of maternal and perinatal outcomes in this population was also assessed. HDP were classified according to Ministry of Health recommendations. Medications were defined using the Anatomical Therapeutic Chemical Classification System and the substance name. In this cohort, 1336 (31.3%) of women had HDP. Gestational hypertension was present in 636 (47.6%) women, 409 (30.6%) had chronic hypertension, 191 (14.3%) pre-eclampsia, and 89 (6.7%) pre-eclampsia superimposed on chronic hypertension. Approximately 70% of women with HDP reported not using any cardiovascular medications. Methyldopa in monotherapy was the most frequent treatment (16%), regardless of the type of HDP. Omega-3 was the medication most frequently reported by women without HDP. Preterm delivery, caesarean section, low birth weight, and neonatal intensive care admissions were more prevalent in women with HDP. Patterns of use of methyldopa were in-line with the Brazilian guidelines as the first-line therapy for HDP. However, the large number of women with HDP not using medications to manage HDP requires further investigation.

## 1. Introduction

Pregnancy complications affect approximately 50 million women in low-and middle-income countries (LMICs) and are associated with severe maternal morbidity and mortality. The most common causes of mortality in LMICs are hemorrhage, hypertension, and maternal infection with 150 deaths per 10,000 women, 20 times higher when compared to high-income countries [1,2]. For every woman who dies of pregnancy-related causes, 20 others experience acute or chronic morbidity with long-term consequences [3].

Hypertensive disorders of pregnancy account for approximately 22% of all maternal deaths in Latin America and the Caribbean [2]. Undiagnosed and untreated hypertension during pregnancy can lead to adverse outcomes in mothers and infants, including stroke, cardiovascular disease, increased admissions to the neonatal intensive care, and maternal and infant mortality [4].

Gestational hypertension is defined by two consecutive measures of a systolic blood pressure ≥ 140 mmHg and/or diastolic blood pressure ≥ 90 mmHg [5]. In-hospital or out-patient assessment is recommended for diagnosis and treatment decisions [6]. Chronic hypertension, gestational hypertension, pre-eclampsia, and chronic hypertension with superimposed pre-eclampsia are the most commonly accepted classifications [7,8]. Among these, pre-eclampsia is the main cause of perinatal and neonatal morbidity and mortality, affecting approximately 2–8% of pregnancies [9,10,11].

In LMICs, there is limited potential for standardization of practices to reduce the occurrence of pre-eclampsia [12,13,14]. Additionally, there are gaps and barriers to the prevention, diagnosis, and treatment of hypertensive disorders of pregnancy (HDP) [15,16]. The recommendations for pharmacotherapy management for HDP varies across countries [17,18,19,20], as do their availability [21]. Aspirin is the recommended therapy for pre-eclampsia prevention [10,11]. Nifedipine, labetalol, and methyldopa have been found to reduce the occurrence of maternal adverse events for women with severe hypertension during pregnancy, serving as a viable intervention in low-resource settings [22]. Although, little is known about the use of these therapies in women with or at risk of HDP in most LMICs.

Numerous studies have examined the patterns of use of prescription and over-the-counter (OTC) medications during pregnancy [23,24,25,26,27]. However, no prior research has aimed to understand the patterns of use of cardiovascular drugs and other therapies for HDP, primarily for those living in LMICs. There is, therefore, a need to better understand how HDP are managed in these countries. The 2015 Pelotas Cohort [28] provides a unique opportunity to assess medication use in women with HDP (pre-existing or not) and to understand outcomes observed in pregnancies complicated by hypertension. Therefore, this study aimed to describe the patterns of use of cardiovascular agents and other medications used by women with and without HDP in the Pelotas (Brazil) Birth Cohort Study. We also aimed to describe characteristics of and to compare the patterns of use, by trimester, between both groups. A secondary objective was to describe the prevalence of maternal and perinatal outcomes in this population.

## 2. Materials and Methods

The methods are based on the STROBE guidelines for reporting of observational research [29].

### 2.1. Study Population

We used population-based data from the 2015 Pelotas Birth Cohort in southern Brazil [28]. The 2015 cohort aimed to provide detailed information on time trends on maternal and child health, behaviours, nutrition, development, and medication use [28]. The data collection was divided into three components: antenatal, perinatal, three and 12 months after birth. Three different types of questionnaires were collected based on the gestational age at enrolment. Women enrolled before 16 weeks’ gestation answered the baseline assessment questionnaire and were subsequently contacted at 20 weeks (range 16–24 weeks) to answer the main assessment questionnaire. Women enrolled after 16 weeks’ gestation responded to a combined assessment questionnaire, as well as those enrolled in the perinatal component (mothers of live-born, not identified before). Detailed information regarding data collection is provided in the methodological article [28].

For this study, we restricted to women with data collected in the antenatal and perinatal period which comprised pregnancies with an expected delivery date (EDD) between 15 December 2014 and 19 May 2016. In both sets of questionnaires, there were different types of questions regarding HDP (See Appendix A). Women were excluded if questions regarding HDP were missing (or ignored, meaning that woman decided not to answer), not allowing to identify HDP.

### 2.2. Definition of Hypertensive Disorders of Pregnancy

The classification of HDP most commonly used in Brazil is that adopted by the Brazilian Federation of Gynecology and Obstetrics—FEBRASGO [5], and by the Ministry of Health guide for attention to high- and low-risk prenatal [19,30]. Therefore, the categories of HDP in this study included, (i) Chronic hypertension; (ii) Pre-eclampsia-eclampsia; (iii) Pre-eclampsia superimposed on chronic hypertension; and (iv) Gestational hypertension. Data on blood pressure measures, diagnostic tests (for example proteinuria), or data on hypertension after delivery were not available so HDP was classified based on self-report by women during the interviews. Most women were enrolled before 20 weeks of gestation [28]; however, the majority of interviews were performed after 20 weeks’ gestation. Therefore, the stage of enrolment was not considered as part of the definition of HDP in our study. The criteria for classifying each HDP is detailed in Appendix A.

### 2.3. Exposure Definition

Women were asked if they used any medication during pregnancy. For each medication reported, the following questions were asked to characterize their usage: (i) “In which trimester/s did you use this medication” and (ii) “How many days per week do/did you use it?” for each trimester. Detailed information regarding the overall use of medications in the Pelotas 2015 Cohort is outlined in the study by Lutz et al. [25]. For our study, we selected all medications whose first level of the Anatomical Therapeutic Chemical (ATC) Classification System was C (ATC—Cardiovascular System). We also selected ATC class B (Blood and Blood Forming Organs), excluding all drugs except those classified as B01AC (Platelet aggregation inhibitors excl. heparin) allowing us to evaluate acetylsalicylic acid (AAS), a medication used specifically for HDP. This approach was used in an attempt to capture treatments that could be used to manage hypertensive disorders in this population. We further classified medication according to the substance name. For all exposures, we assumed women were continuously exposed (1 to 90 days) for the entire length of the trimester of reported use.

For the prevalence of use of pharmacological therapies, we further classified medications according to the type of treatment: (i) No cardiovascular drug, if women reported not using any antihypertensive medication, aspirin, or other cardiovascular agents during pregnancy; (ii) Methyldopa monotherapy, if they reported using only methyldopa in any or all trimesters; (iii) Any other cardiovascular drug without methyldopa, if they were using any other drug and methyldopa was not part of the therapy; and (iv) Any other cardiovascular drug or aspirin plus methyldopa, if they reported using methyldopa in combination with any other cardiovascular drug. The findings were stratified by type of treatment and type of HDP. We used this categorization since it followed the current guideline recommendations in Brazil regarding the use of medications for women with hypertension during pregnancy [5,19] (See Appendix A).

### 2.4. Baseline Characteristics

Characteristics of women enrolled in the Pelotas Cohort were collected at the baseline interview according to the component (antenatal or perinatal). We included maternal age (years), ethnicity (white, black, other), mother’s education (number of completed years of study categorized into four groups: 0–8, 9–11 and ≥12 years), and ABEP levels were used to define socioeconomic status. ABEP is the socioeconomic index consisting of a series of questions about the possession of durable goods and the educational level of the head of the household, in which A represents the highest level, while E is the lowest one. Smoking was defined as tobacco use in different stages of pregnancy and further classified as a binary yes/no variable if women answered yes to use of tobacco at least once during the follow-up period. The same rationale was used for classifying alcohol use and other illicit drugs. Multifetal gestation was used to define women with multiple pregnancies. Prepregnancy body mass index (BMI) was calculated based on self-reported height and weight and further categorized as <18.5, 18.5–24.9, 25.0–29.9, ≥30.0 kg/m^2^. Gestational weight gain was defined as the weight recorded in the National Pregnancy Card. It was computed as the weight before becoming pregnant minus the weight at the end of pregnancy (self-reported) or the weight recorded at the last medical consultation. Hospital admissions were defined as those occurring during pregnancy (yes/no) and comorbidities and conditions of pregnancy were defined based on the questions regarding the women’s condition (yes/no) in the antenatal and perinatal components. We included the following conditions from the questionnaires: pre-existing renal disease, diabetes mellitus, depression or other nervous disorders, urinary infections, thyroid disorders, and heart disease.

### 2.5. Maternal and Perinatal Outcomes

Maternal and perinatal outcomes included preterm birth defined as a birth < 37 weeks gestation (yes/no), mode of delivery (vaginal or caesarean section), admissions to the neonatal intensive care unit (NICU; yes/no), and low birth weight defined as birth weight < 2500 g (yes/no).

All deliveries were performed in one of five Pelotas maternities (two medical school hospitals, one of them exclusively for users of the Unified Health System—SUS).

### 2.6. Statistical Analysis

We conducted descriptive analyses to assess the demographic, behaviour, and clinical characteristics of our study population. Results are presented as frequencies for categorical variables and mean and standard deviations for continuous variables. Prevalence of use of cardiovascular agents is reported according to the type of treatment by type of HDP. We also described all medications used by ATC/substance across gestation until the delivery (first, second, and third trimester). The number of cardiovascular medications used, including AAS, is presented by trimester of use. All statistical analyses were conducted using SAS version 9.4 (SAS Institute, Cary, NC, USA).

### 2.7. Ethics Approval

This study was approved by the *Universidade Federal de Pelotas* School of Physical Education Ethics Committee (522.064) and was registered in the National Ministry of Health’s *Plataforma Brasil*. All mothers signed a free and informed consent form before being interviewed.

## 3. Results

Of the 4270 eligible women in the original cohort, 4262 were included in our study. The majority of antenatal or perinatal questionnaires were answered after 20 weeks’ gestation (85.4%; 3641/4262). Eight women were excluded due to incomplete data on HDP (see Appendix A). HDP was present in 1336 (31.4%) of women in the cohort (Figure 1).

Table 1 shows the characteristics of women with and without HDP in the Pelotas cohort. Among women with HDP, 636 (48%) reported having gestational hypertension, 409 (31%) chronic hypertension, 191 (14%) pre-eclampsia, and 89 (7%) pre-eclampsia superimposed on chronic hypertension. The majority of women with HDP were white (66.5%), completed 9–11 years of education (38.2%), and were of lower-middle socioeconomic class (54.1%, ABEP level C). The prevalence of prepregnancy obesity (BMI ≥ 30 kg/m^2^) in women with HDP was almost 3 times the prevalence of women without HDP. No differences were found in gestational weight gain between pregnancies affected and not affected by HDP. Comorbidities and complications of pregnancy were slightly higher in women with HDP and they were also more likely to be hospitalized (30.2%). The most prevalent diseases in this population were urinary infection (45%), followed by depression (12%) and gestational diabetes (8%), with the latter doubled in women with HDP.

Preterm delivery and caesarean section were more common in women with HDP (72% and 61% respectively). Women with HDP were also more likely to give birth to babies with low birth weight and to report more NICU admissions (10% and 8%, respectively).

The types of cardiovascular treatments used by type of HDP are presented in Table 2. Approximately 67% of women with HDP reported not using any cardiovascular medication during pregnancy. For those using a pharmacological treatment, methyldopa in monotherapy was the treatment most frequently reported, regardless of the type of HDP. Approximately 9% of women reported using cardiovascular therapy without a diagnosis of HDP. The number of women with missing information on medication use ranged from 3 to 18% by type of HDP (See Appendix A).

When examined by trimester, the majority of women with gestational or chronic hypertension received no treatment or monotherapy. Figure 2 shows that the non-use of cardiovascular medications decreased across gestation among women with both gestational and chronic hypertension. Monotherapy increased, while the frequency of use of 2 or more drugs for cardiovascular conditions among women with gestational and chronic hypertension was low and did not increase across gestation (Figure 2).

When examining the substance type by trimester of use, we found that women with HDP were more likely to increase methyldopa use over the pregnancy with prevalence increasing from 11% to 19% from the first to third trimester (Figure 3A and Appendix A). This was in conjunction with reduced use of all other classes of medication throughout their pregnancy, except for omega-3 that was the third most commonly reported medication (0.9 to 2.0% across pregnancy). Acetylsalicylic acid (AAS) was the second most commonly reported medication used, with the frequency increasing in the second trimester and subsequently decreasing in the third trimester (2.4%, 3.5%, and 2.4%, respectively). Women who did not report HDP were found to use cardiovascular agents during pregnancy. Among these women, we found that the use of “Other lipid modifying agents” increased from 1.1 to 4.5% and isoxsuprine increased from 0.4 to 1.2% throughout pregnancy. We also found AAS use among women without HDP, but its use was lower and decreased throughout pregnancy when compared to women with HDP (Figure 3B) (see all cardiovascular drugs and frequencies in Appendix A).

## 4. Discussion

Our findings show a high prevalence of HDP in the Pelotas birth cohort, with three out of ten pregnant women reporting an HDP. We also found that these women have more comorbidities and hospital admissions, which represents an important burden for this population. The prevalence of pre-eclampsia and eclampsia in this study was higher than reported in other LMICs [31], affecting 6% of pregnancies. Interestingly, we found a low prevalence of use of treatments for managing cardiovascular diseases among women with HDP. Methyldopa was the most commonly reported antihypertensive medication, which is in-line with the current Brazilian recommendations. Infrequent use of other classes of cardiovascular drugs was observed, although medications contraindicated during pregnancy were reported. We also found a high frequency of omega-3 use among women who did not report an HDP. Management of cardiovascular drugs and AAS by both women affected and not affected by HDP requires further attention and needs to be better understood in light of the morbidity of women and infants in this population.

In Brazil, maternal health programs date from the 1980s with the National Women’s Health Program in 1984, the National Women’s Health Program National Women’s Health Program of Pregnancy and Childbirth in 2000, and the Pact for the Reduction of Maternal and Newborn Mortality in 2004 [32]. Initiatives of the Ministry of Health are underway, but measures related to HDP require further attention. Few differences exist in terms of diagnosis and therapy management compared to high resources countries, although, like in other LMICs the problem seems to be the implementation of good-practices [16]. The current guidelines for the management of pregnancies in primary care have not been updated in over five years [19,30]. Models for evidence-based guidelines, implementation, and surveillance exist [4,33]. However, barriers to the adoption of such practices in Brazil exist.

Based on our findings, approximately 20% of pregnant women had at least one hospital admission, of which 30% were due to HDP. Although self-reported measures of admission may introduce the potential for misclassification, it is worth highlighting the potential importance and impact of this elevated rate of hospital admissions in this population and the impact on the short- and long-term health of women. Although close monitoring of pregnancies affected by or at high risk of HDP is recommended, the extent at which these women are being monitored and managed is not well understood, considering the high rates of mortality in Brazil and other LMICs. No surveillance data exist for monitoring and reducing the impact of HDP in these countries, and the burden of families affected by these conditions [34].

As expected, the percentage of caesarean section, preterm birth (PTB), low birth weight (LBW), and neonatal intensive care unit (NICU) admission were higher for women with HDP [31]. Trends in LBW and PTB were previously reported in four population-based birth cohorts in Pelotas, Brazil (1982–2015) [35]. Despite differences in the definition of gestational age between cohorts, an increase in preterm deliveries in Pelotas was consistent across all cohorts. Furthermore, contrary to what was expected, a reduction in PTB and an increase of LBW due to improvements in the quality of care were not found. Inequalities in the prevalence of caesarean section may be explained by the higher prevalence among women of higher socioeconomic status (SES). Our study suggests that SES and prevalence of HDP may also explain the high rates of caesarean section. The only effective treatment for complications due to HDP and mainly severe pre-eclampsia and eclampsia is delivery [10,19]. The choice of mode of delivery is carefully evaluated based on the gestational age and foetal factors [19]. Delivery is typically best for the mother but not always good for foetuses, leading to increased prevalence of preterm birth, LBW, and NICU admissions [7], a tendency observed in our study.

This study is the first to describe the prevalence of the use of cardiovascular treatments and AAS among women with HDP in Brazil. Despite the lack of information relating to the onset of HDP, blood pressure measures, as well as the date of treatment initiation and hospital admissions, this study highlights the need to better understand current practices surrounding the treatment and management of women with HDP. The current guidelines recommend measuring blood pressure at all prenatal consultations, with hypertensive changes associated with serious foetal complications and maternal and perinatal mortality [30]. Women with chronic arterial hypertension, either moderate or severe, or taking antihypertensive medications are classified as having a high-risk pregnancy and referral for specialized service is required [5,30]. According to the Ministry of Health guidelines, women with controlled chronic hypertension, with no sign of disease in target organs, will rarely need drug therapy. If needed, the same agents used before pregnancy can be maintained, except for angiotensin-converting enzyme inhibitors and angiotensin II receptor blockers, for which the guidelines recommend discontinuing. For those with hypertension diagnosed after 20 weeks, no antihypertensive treatment is recommended unless blood pressure measures cross clinical thresholds for a systolic blood pressure ≥ 150 mmHg and diastolic blood pressure from 100 to 110 mmHg or signs of target organ damage. Based on these recommendations, we assume that most women in our study had well-controlled chronic hypertension and mild gestational hypertension, without target organ damage and, consequently, did not require antihypertensive medication in the first half of pregnancy. If this is indeed the case, we would expect a low prevalence of hospital admissions in women with HDP, as well as a low frequency of pre-eclampsia/eclampsia, which was not the case. A recent study found an increased risk of serious maternal complications when less-tight (versus tight) control of hypertension was adopted after 28 weeks [36], which could be a plausible hypothesis for our findings. Women in our cohort would be starting stricter pressure control late, which would increase their morbidity. However, further studies will need to be conducted to confirm this hypothesis.

Concerning the use of angiotensin-converting enzyme inhibitors and angiotensin II receptor blockers observed in women with HDP, it is not unexpected primarily in the first trimester of pregnancy, in which many women are not yet aware that they are pregnant [37]. Although evidence suggests that women should discontinue or change therapies as soon as they become pregnant [18,19], the evidence is still unclear whether exposure in the second and third trimester increases the risks of adverse outcomes [38]. This could therefore explain use of these drugs in women later in gestation.

Our study highlights an important consideration for the use of AAS and omega-3s. The proportion of women taking AAS was low. However, we expect that at minimum women with chronic hypertension were using this medication, considering the *FEBRASGO* recommendation [5]. If AAS is an effective prevention for PE, this simple and cost-effective treatment should be considered a potential strategy for LMICs. Our study also showed that women with and without HDP reported using omega-3s during pregnancy, with a higher frequency among women with HDP. Clinical guidelines do not recommend nutritional supplements for HDP prevention due to the low-quality of available evidence [39]. Omega-3s, however, have been extensively studied and a recent systematic review demonstrated that omega-3s may reduce the risk of pre-eclampsia and PTB. Although, the evidence still includes studies of low quality [40]. Therefore, further studies are needed to understand if omega-3s are indeed associated with blood pressure control and whether physicians are prescribing this medication as prophylactic treatment. Isoxsuprine is a peripheral vasodilator used as a tocolytic agent, prolonging pregnancy in women at risk of abortion or premature delivery [41]. In our study, women with and without HDP reported using it. The Ministry of Health guidelines recommend the use of tocolytics for premature labour, but no recommendation for using isoxsuprine was found [19], which will need further investigation.

Our study had several limitations that should be addressed. First, there is the potential for misclassification of HDP as a result of self-reported HDP. Data on diagnoses or blood pressure measures within this population were not available and HDP were classified according to answers related to pregestational and perinatal morbidities. Logistics for collecting clinical information and recording blood pressure would probably make this type of study unfeasible. Therefore, self-reported information usually is the only feasible way for conducting large-scale studies in low-resource settings. Validation studies are thus recommended to ensure that self-reported information can be used. However, no validation studies were conducted for HDP diagnosis in this cohort. A national cross-sectional, multicenter, hospital-based study in Brazil has demonstrated moderate accuracy in self-reported HDP by pregnant women [42]. The authors observed that pregnant women overreported HDP (15.9%), when compared to medical records and antenatal cards (11.1%). Second, the frequencies for overall HDP, as well as for the subtypes of HDP, were higher than estimates usually reported for this type of population. Misclassification and overreporting may be a reasonable explanation. However, these may represent accurate measures of HDP in this population, considering the Pelotas Cohort enrolled almost all pregnant women in this city, which suggests that these figures cannot be explained by selection bias. We recognize the limitations on estimating any association using these data, but the proportions call for attention on how pregnancies, affected or at risk of HDP, are managed. These conditions are not recorded as a basic cause of death, not reflecting the real importance and reality of its frequency [43]. Mortality due to HDP is historically underreported in Brazil and Maternal Mortality Ratio due to maternal hypertensives disorders was 13 maternal deaths per 100,000 live births in 2019, suggesting that prevention strategies concerning HDP must be re-evaluated [44]. Finally, there is also the potential for recall bias concerning the use of medications. Although, a study by Lutz et al. [25] suggests that this is minimized by the classification of medications based on the information at the antenatal visit.

## 5. Conclusions

This study highlights the need for further studies concerning the treatment and management of HDP. Cohorts with validated diagnoses, timing and onset of diagnoses, and timing of prescriptions would allow for the assessment of the effectiveness of antihypertensive medications and other drugs for the management of HDP in this population. Despite methyldopa treatment being in-line with Brazilian guidelines, a large number of women with HDP not being prescribed antihypertensive agents requires further investigation. The estimates of HDP, morbidity, and maternal and perinatal outcomes require further attention in this population.

## Figures and Tables

**Figure 1 ijerph-17-08541-f001:**
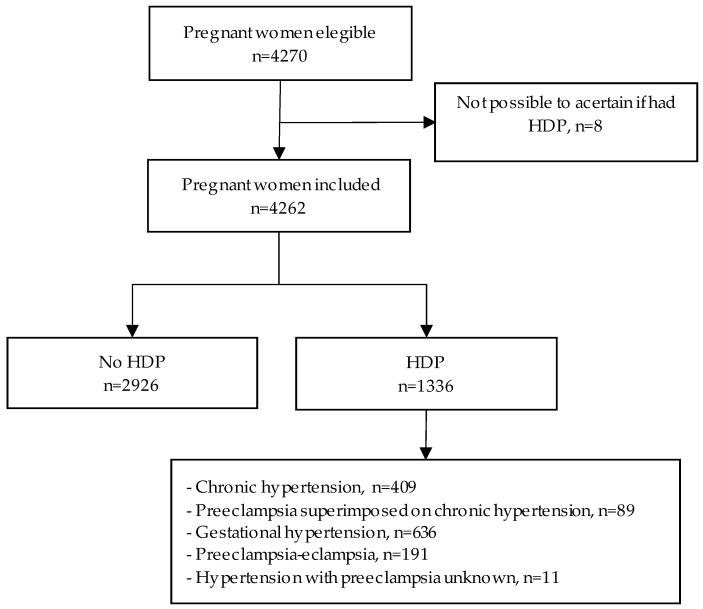
Flowchart of women included in the study. Abbreviation: HDP: Hypertensive disorder of pregnancy.

**Figure 2 ijerph-17-08541-f002:**
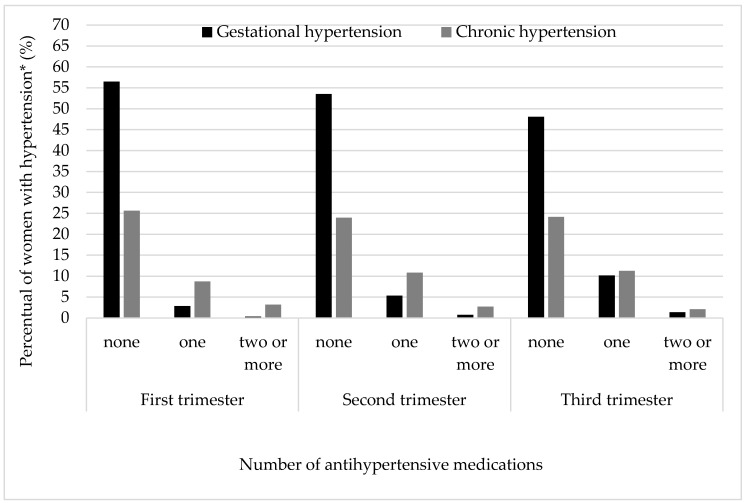
Number of cardiovascular medications taken by trimester by women with gestational and chronic hypertension (*n* = 1285). 2015 Pelotas birth cohort. * Missing or ignored on medication use *n* = 51.

**Figure 3 ijerph-17-08541-f003:**
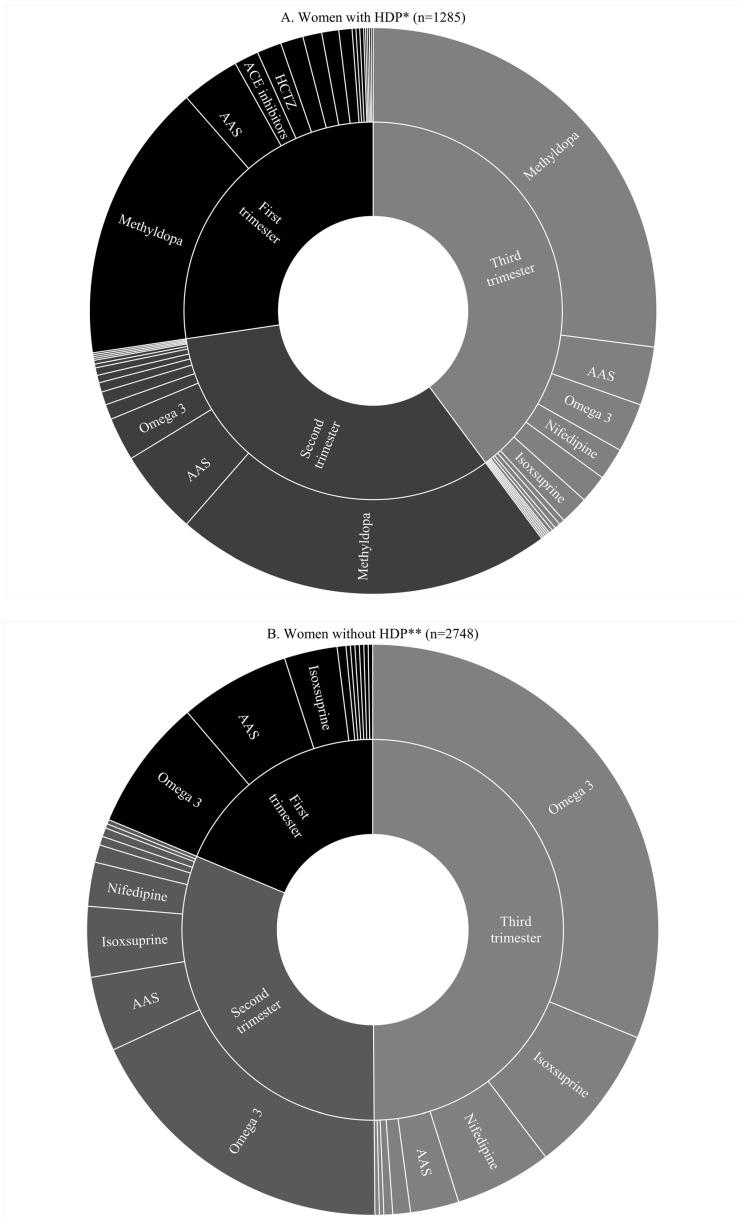
(**A**) Use of cardiovascular medications and AAS in women with HDP by trimester (*n* = 1285). (**B**) Use of cardiovascular medications and AAS in women without HDP by trimester (*n* = 2748). * Missing or ignored on medication use *n* = 51. ** Missing or ignored on medication use *n* = 178. Abbreviations: HDP: Hypertensive Disorders of Pregnancy; AAS: Acetylsalicylic acid; HCTZ: Hydrochlorothiazide; ACE inhibitors: Angiotensin-converting-enzyme inhibitors (captopril or enalapril).

**Table 1 ijerph-17-08541-t001:** Baseline characteristics of women in the Pelotas cohort 2015, according to hypertensive disorders of pregnancy (HDP) (*n* = 4262) *.

Characteristic	Total	Non-HDP	HDP
	(*n* = 4262)	(*n* = 2926)	(*n* = 1336)
Age, mean (SD) [years]	27.6 (6.6)	27.4 (6.5)	28.0 (6.8)
Ethnicity, *n* (%)			
White	2999 (70.4)	2111 (72.1)	888 (66.5)
Black	679 (15.9)	423 (14.5)	256 (19.2)
Other or ignored	578 (13.6)	387 (13.2)	191 (14.3)
Missing or not answered	6 (0.1)	5 (0.2)	1 (0.1)
Mother’s Education (years complete of schooling), *n* (%)			
0–8 years	1486 (34.9)	983 (33.6)	503 (37.6)
9–11 years	1462 (34.3)	951 (32.5)	511 (38.2)
12 years or more	1314 (30.8)	992 (33.9)	322 (24.1)
Socioeconomic index (ABEP 3 levels) ^a^, *n* (%)			
A–B	1257 (29.5)	944 (32.3)	313 (23.4)
C	2049 (48.1)	1326 (45.3)	723 (54.1)
D–E	811 (19.0)	548 (18.7)	263 (19.7)
Missing or not answered	145 (3.4)	108 (3.7)	37 (2.8)
Smoking, *n* (%) ^b^	875 (20.5)	586 (20.0)	289 (21.6)
Alcohol, *n* (%) ^b^	1543 (36.2)	1057 (36.1)	486 (36.4)
Illicit drug use during pregnancy (other than alcohol), *n* (%) ^b^	35 (0.8)	24 (0.8)	11 (0.8)
Multifetal gestation, *n* (%) (twins)	48 (1.1)	30 (1.0)	18 (1.3)
Prepregnancy BMI, *n* (%) (kg/m^2^)			
<18.5	155 (3.6)	133 (5.5)	22 (1.6)
18.5–24.9	2036 (47.8)	1590 (54.3)	446 (33.4)
25–29.9	1158 (27.2)	756 (25.8)	402 (30.1)
≥30	777 (18.2)	353 (12.1)	424 (31.7)
Missing or not answered	136 (3.2)	94 (3.2)	42 (3.1)
Gestational weight gain, mean (SD) (kg)	11.74 (6.6)	11.65 (6.3)	11.95 (7.2)
Hospital admissions, *n* (%)	837 (19.6)	434 (14.8)	403 (30.2)
Comorbidities, *n* (%)			
Pre-existing renal disease	204 (4.8)	122 (4.2)	82 (6.1)
Heart disease	58 (1.4)	32 (1.1)	26 (1.9)
Gestational diabetes mellitus	363 (8.5)	187 (6.4)	176 (13.2)
Depression or other nervous disorders	503 (11.8)	303 (10.4)	200 (15.0)
Urinary infection	1911 (44.8)	1262 (43.1)	649 (48.6)
Hypothyroidism/thyroid disease	316 (7.4)	206 (7.0)	110 (8.2)
Use of progesterone, evocanil, duphaston or utrogestan, *n* (%)	533 (12.5)	391 (13.4)	142 (10.6)
Gestational age, mean (SD) [days]	269.0 (17.2)	270.0 (16.8)	266.8 (18.1)
Maternal and perinatal outcomes, *n* (%)			
Pre-eclampsia or eclampsia	280 (6.6)	-	280 (21.0)
Pre-term delivery < 37 weeks	653 (15.3)	396 (13.5)	257 (19.2)
Caesarean section	2757 (64.7)	1799 (61.5)	958 (71.7)
Low birth weight (<2500 g)	384 (9.0)	243 (8.3)	141 (10.5)
NICU admission	281 (6.6)	175 (6.0)	106 (7.9)

Abbreviation: SD: standard deviation; HDP: hypertensive disorder of pregnancy; BMI: Body mass index; NICU: Neonatal Intensive Care Unit. * Number of women assessed, after exclusions. ^a^ ABEP = socioeconomical status classified according to *Critério de Classificação Econômica Brasil* 2013 (CCEB 2013—Brazilian Economic Classification Criterion) of *Associação Brasileira de Empresas de Pesquisa* (ABEP—Brazilian Association of Survey Companies). Available from: http://www.abep.org. It is an index consisting of a series of questions about the possession of durable goods and the educational level of the head of the household, in which A represents the highest level, while E is the lowest one. ^b^ Use of tobacco, alcohol, and illicit drugs, at least once, any time of pregnancy.

**Table 2 ijerph-17-08541-t002:** Characteristics of cardiovascular therapy during pregnancy in women in the Pelotas cohort 2015, according to type of hypertensive disorder of pregnancy (HDP). (*n* = 4033) *.

Characteristic	Total	No Cardiovascular Drug **	Methyldopa Monotherapy	Any Other Cardiovascular Drug without Methyldopa **	Any Other Cardiovascular Drug + Methyldopa **
	(*n* = 4033)	(*n* = 3374)	(*n* = 218)	(*n* = 357)	(*n* = 84)
**No HDP, *n* (%)**	2748 (68.1)	2482 (90.3)	1 (0.0)	264 (9.6)	1 (0.0)
**Type of HDP ***, *n* (%)**					
Chronic hypertension	393 (9.7)	235 (59.8)	83 (21.1)	30 (7.6)	45 (11.4)
Gestational hypertension	613 (15.2)	493 (80.4)	60 (9.8)	45 (7.3)	15 (2.4)
Pre-eclampsia superimposed on chronic hypertension	84 (2.1)	29 (34.5)	34 (40.5)	7 (8.3)	14 (16.7)
Pre-eclampsia-eclampsia	186 (4.6)	132 (71.0)	35 (18.8)	11 (5.9)	8 (4.3)
Hypertension during pregnancy with pre-eclampsia unknown	9 (0.2)	3 (33.3)	5 (55.6)	0	1 (11.1)

* Number of women assessed, after exclusions. There were 229 observations in which medication use was missing or classified as ignored (not answered). ** Including B01AC (Platelet aggregation inhibitors excl. heparin), represented by acetylsalicylic acid (AAS). *** Adapted according to recommended by the FEBRASGO and Ministry of Health guide; 9 (0.2%) hypertension cases were not possible to classify because there was no information about pre-eclampsia/eclampsia.

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
