# Peer review of "Hypertensive Disorders of Pregnancy and Medication Use in the 2015 Pelotas (Brazil) Birth Cohort Study"

_ijerph, 2020, doi:10.3390/ijerph17228541_

Round 1
Reviewer 1 Report
I value the paper and its content, and have provided specific comments or suggestions on aspects of the paper. Interesting factual observation in a cohort with an overall high incidence of hypertension, and with still about 70% untreated.
specific comments
abstract:
140,000 as absolute number, but the % should be added; not sure on the add on value of omega-3 as most commonly used medication in non HDP cases ?
introduction: the statement on aspirin is supported by ref 10 (check ref, title missing), but is it also in the national guidelines ?
table 1: drug use during pregnancy, i assume that this refers to illicit drugs ? is there any relevance to furhter discuss the differences in 'pre'pregnancy characteristics between HDP vs no HDP cases ?
(other typo: progesterone, and ou should read or)
figure 2 and 3: can you further improve the quality of the figure ?
the presence of ACE inhibitors in the first trimester of pregnancy should be furhter stressed.
are the any data on placenta abruption ?
the omega-3 statements should be further tuned down as this in an association, not corrected for covariates or risk factors.
Author Response
abstract:
140,000 as absolute number, but the % should be added; not sure on the add on value of omega-3 as most commonly used medication in non HDP cases ?
Response:
We appreciate all reviewer’s considerations and the corrections were made accordingly.
Concerning the statement ‘140,000 as absolute number, but the % should be added’, we have added the proportion as per the reviewer’s request.
The correction was made and can be found on page 1, line 15. The text is as follows:
Abstract: Hypertensive disorders of pregnancy account for approximately 22% of all maternal deaths in Latin America and the Caribbean. Pharmacotherapies play an important role in preventing and reducing the occurrence of adverse outcomes.
Regarding omega-3 use, there is little evidence to support the use of omega-3 use as a preventive measure for HDP, though a higher proportion of use among women without HDP was found in our study. Omega-3s have been extensively studied and a recent systematic review demonstrated that omega-3s may reduce the risk of preeclampsia and PTB (1). However, the studies included had low quality and further research is needed to understand if omega-3s are associated with blood pressure control and whether physicians are prescribing this medication as a prophylactic treatment for women at risk of HDP.
- Middleton, J.C. Gomersall, J.F. Gould, E. Shepherd, S.F. Olsen, M. Makrides, Omega‐3 fatty acid addition during pregnancy, Cochrane Database of Systematic Reviews (11) (2018).
We have kept the statement concerning omega-3 in the abstract and the result is discussed on page 11, lines 318-322.
Omega-3s, however, have been extensively studied and a recent systematic review demonstrated that omega-3s may reduce the risk of preeclampsia and PTB. Although, the evidence still includes studies of low quality (40). Therefore, further studies are needed to understand if omega-3s are indeed associated with blood pressure control and whether physicians are prescribing this medication as prophylactic treatment.
The questions are what were the reasons for using it? Were they at risk for developing HDP, or if they must be treated for HDP with recommended medication instead of using omega-3?
introduction: the statement on aspirin is supported by ref 10 (check ref, title missing), but is it also in the national guidelines ?
Response:
We have revised reference 10 to include the title as per the reviewer’s comment.
This is the reference corrected:
- Mol BWJ, Roberts CT, Thangaratinam S, Magee LA, de Groot CJM, Hofmeyr GJ. Pre-eclampsia. Lancet. 2016 Mar 5;387(10022):999-1011.
The National guidelines recommend the use of aspirin for patients with a diagnosis of thrombophilia. However, the guideline from the Federação Brasileira das Associações de Ginecologia e Obstetrícia (FEBRASGO) recommends the use of aspirin for preventing preeclampsia in high-risk patients. This has been included in the text on page 2, line 73. The text corrected can see below:
Aspirin is the recommended therapy for preeclampsia prevention (10, 11).
- Federação Brasileira das Associações de Ginecologia e Obstetrícia (FEBRASGO). Pré-eclâmpsia nos seus diversos aspectos. In: FEBRASGO. SOeR, editor. São Paulo2017. p. 56.
table 1: drug use during pregnancy, i assume that this refers to illicit drugs? is there any relevance to furhter discuss the differences in 'pre'pregnancy characteristics between HDP vs no HDP cases?
Response:
The text has been revised to clarify the use of illicit drugs on page 3, line 135 of the manuscript and on Table 1, page 6, as follows:
Illicit drug use during pregnancy (other than alcohol), n (%)
Concerning the relevance of discussing pre-pregnancy characteristics among women with HDP and without HDP, our objective was to present the characteristics that are considered for referring pregnant women to high-risk prenatal care. According to the Ministry of Health guideline, factors as higher pre-pregnancy BMI, excessive gestational weight gain, pre-existing renal disease, gestational diabetes, among others, are reasons for evaluating the referral and are indicative of the development of HDP. In our study, we showed that women with HDP were more likely to report a higher pre-pregnancy BMI, as well as renal disease than those without HDP, that might reinforce the conduct described in current guidelines. Further studies are needed to understand if women were properly managed, considering their pre-existing conditions.
(other typo: progesterone, and ou should read or)
Response:
We thank the reviewer for identifying a typo, we have revised the text on page 6, Table 1.
Use of progesterone, evocanil, duphaston or utrogestan, n (%)
figure 2 and 3: can you further improve the quality of the figure?
Response:
The figures have been improved as per the reviewers’ request.
the presence of ACE inhibitors in the first trimester of pregnancy should be furhter stressed.
Response:
A discussion of the use of ACE inhibitors was added to page 11, lines 305-310 of the manuscript. The new paragraph is shown below:
Concerning the use of angiotensin-converting enzyme inhibitors and angiotensin II receptor blockers observed in women with HDP, it is not unexpected primarily in the first trimester of pregnancy, in which many women are not yet aware that they are pregnant (37). Although evidence suggests that women should discontinue or change therapies, as soon as they become pregnant (18, 19) the evidence is still unclear whether exposure in the second and third trimester increases the risks of adverse outcomes (38). This could therefore explain the use of these drugs in women later in gestation.
are the any data on placenta abruption?
Response:
Unfortunately, data on placental abruption was not available in this cohort.
the omega-3 statements should be further tuned down as this in an association, not corrected for covariates or risk factors.
Response:
Our study was descriptive in nature and was not intended to demonstrate associations. The results presented in the manuscript are frequencies or proportion of use of omega-3’s among women with HDP. Our discussion does not intend to support any association. According to the text below, we discuss a possible association showed in a systematic review, in which decreasing risk for preeclampsia and pre-term birth was observed. However, the studies had low quality and do not support using it for preventing HDP or other adverse outcomes.
Omega-3s, however, have been extensively studied and a recent systematic review demonstrated that omega-3s may reduce the risk of preeclampsia and PTB. Although, the evidence still includes studies of low quality (40). Therefore, further studies are needed to understand if omega-3s are indeed associated with blood pressure control and whether physicians are prescribing this medication as prophylactic treatment.
Further quality research is needed to answer the role of omega-3 in pregnancies at risk of HDP.
Reviewer 2 Report
Review for IJERPH : Use of antihypertensive medication among women with hypertensive disorders of pregnancy in the 2015 Pelotas (Brazil) birth cohort study
- Summarize the manuscript's content: This is a retrospective review of a large questionnaire based study performed in the city of Pelotas Brazil in 2015 among 4262 women of whom 1336 had some form of hypertension. The authors used patient completed questionnaires, which were completed by patients antenatal, around the time of delivery, and postpartum at 3 and 12 months. Not all women filled out all the questionnaires. These were completed at the time of study entry. The authors describe differences or similarities between women diagnosed with any type of hypertensive disease and those without. Approximately 30% of women were diagnosed with some form of hypertension in pregnancy. They then described patterns of medication usage, most importantly antihypertensive medications among women with hypertension. Usage was classified as none, CHTN, GHTN preeclampsia and CHTN with superimposed preeclampsia. Use was reported by trimester if any medication was used during that trimester. Usage was also classified according to whether AMD or alternative medications were used. The authors report that the Ministry of Health supports AMD as the first line medication for women with BP values equal to or greater than 150 systolic, 100 diastolic or with evidence of end organ damage. The significant findings for BP medications are that a majority did report use of any medication, the most commonly used medication in pregnancy was AMD, medication use increased during pregnancy for women with CHTN as well as for women with GHTN/preeclampsia. The authors conclude that more information is needed, and would like to recommend studies into why so few women used antihypertensive medication.
- The strengths of the manuscript are the large population based study and descriptive study regarding management of hypertension in pregnancy in this region of Brazil. The patient recall is within a short period so that it is likely to be better than long-term recall from previous pregnancies.
- The perceived limitations are:
- This is a retrospective study based on a diagnosis that is ascertained by the women who were enrolled. There is no additional form of ascertainment or validation of the diagnosis, a limitation recognized by the authors. It is unknown how many women were enrolled prior to 20 weeks, when the diagnosis of CHTN might have been better recalled, and how many were enrolled after 20 weeks when the diagnosis of GHTN would have been more likely to be made. It would help to report how many women were enrolled at each stage of the pregnancy.
- The proportion of women with the various forms of hypertensive disease varies and is not concordant with what is often reported in the literature. The overall frequency of HDP of 30% is rather high and is concerning for selection bias. It would benefit the manuscript to comment further on this issue. There is a 6.7% incidence of preeclampsia, which is high, but within the parameters reported in the literature. The prevalence of CHTN is 10% and for GHTN is 15% which also raises the question of validation of the diagnosis. The authors might want to comment further on this.
- The structure of the manuscript compares use of medication among women with and without hypertension, but the purpose of the paper is to analyze medication usage patterns among hypertensive women. Figure 3B does not contribute anything to the manuscript as it reports on medications that are no antihypertensive and does not contribute to the overall understanding of the topic being discussed.
- Among the antihypertensive medications included in Figure 3A are isoxuprine, low dose aspirin and Omega 3 fatty acid supplements, none of which is an antihypertensive medication.
- Clinical relevance as a descriptive study reporting on medication patterns by hypertensive women in one city of Brazil. The patterns of medication use are not necessarily generalizable to other settings were AMD is not a first line drug. If participant outcomes including modes of delivery and complications are reported, in addition to short term neonatal outcomes, this could contribute significantly to supporting use of AMD compared with other commonly used medications that lead to a more efficient response in management of HDP such as nifedipine.
- Lines 260-273 of the discussion are not relevant to this manuscript.
Author Response
Review for IJERPH : Use of antihypertensive medication among women with hypertensive disorders of pregnancy in the 2015 Pelotas (Brazil) birth cohort study
- Summarize the manuscript's content: This is a retrospective review of a large questionnaire based study performed in the city of Pelotas Brazil in 2015 among 4262 women of whom 1336 had some form of hypertension. The authors used patient completed questionnaires, which were completed by patients antenatal, around the time of delivery, and postpartum at 3 and 12 months. Not all women filled out all the questionnaires. These were completed at the time of study entry. The authors describe differences or similarities between women diagnosed with any type of hypertensive disease and those without. Approximately 30% of women were diagnosed with some form of hypertension in pregnancy. They then described patterns of medication usage, most importantly antihypertensive medications among women with hypertension. Usage was classified as none, CHTN, GHTN preeclampsia and CHTN with superimposed preeclampsia. Use was reported by trimester if any medication was used during that trimester. Usage was also classified according to whether AMD or alternative medications were used. The authors report that the Ministry of Health supports AMD as the first line medication for women with BP values equal to or greater than 150 systolic, 100 diastolic or with evidence of end organ damage. The significant findings for BP medications are that a majority did report use of any medication, the most commonly used medication in pregnancy was AMD, medication use increased during pregnancy for women with CHTN as well as for women with GHTN/preeclampsia. The authors conclude that more information is needed, and would like to recommend studies into why so few women used antihypertensive medication.
- The strengths of the manuscript are the large population based study and descriptive study regarding management of hypertension in pregnancy in this region of Brazil. The patient recall is within a short period so that it is likely to be better than long-term recall from previous pregnancies.
- The perceived limitations are:
- This is a retrospective study based on a diagnosis that is ascertained by the women who were enrolled. There is no additional form of ascertainment or validation of the diagnosis, a limitation recognized by the authors. It is unknown how many women were enrolled prior to 20 weeks, when the diagnosis of CHTN might have been better recalled, and how many were enrolled after 20 weeks when the diagnosis of GHTN would have been more likely to be made. It would help to report how many women were enrolled at each stage of the pregnancy.
Response:
In this study, approximately 50% of women were enrolled before 20 weeks of gestation. For the cohort used in this study, 48.9% of women were enrolled before 20 weeks. The response rate for the questionnaires on pre-pregnancy comorbidities was answered by only 14.6% of women prior to 20 weeks of pregnancy. Our variables and algorithm for classifying HDP are described in the supplemental material, pages 1-2, Appendix S1-S2, according the table below.
Appendix S1.
|
Table 1. Variables used for hypertensive disorders of pregnancy (HDP) definition. |
||
|
Questionnaires |
Block |
Questions related to hypertension (0=No; 1=Yes; 9=IGN) |
|
Prenatal follow-up - Full interview (antenatal) |
PREGESTIONAL MORBITIES |
|
|
Before this pregnancy you had or had: (Q46) High blood pressure or hypertension |
||
|
And now, during this pregnancy, so far, you have presented any of these health problems: (Q61a.) Hypertension |
||
|
Perinatal |
||
|
PRENATAL BLOCK AND GESTATIONAL MORBIDITY |
||
|
Now let's talk about some diseases that may have occurred during pregnancy. During the pregnancy.... (Q106). Did you have high blood pressure? IF NOT OR IGN go to 108 |
||
|
(Q107). Did you have high blood pressure before pregnancy? |
||
|
|
|
(Q108). Did you have eclampsia or pre-eclampsia? |
|
Full questionnaires available in Portuguese. URL: http://epidemio-ufpel.org.br/site/content/coorte_2015-en/questionnaires.php |
||
Appendix S2.
HDP classification:
- Chronic hypertension was attributed to those who answered yes to questions about hypertension before the current pregnancy in either the antenatal or perinatal components of the questionnaires.
- Those who had chronic hypertension and answered yes to the question of preeclampsia in the perinatal component were classified as having preeclampsia superimposed on chronic hypertension.
- Women who answered yes to hypertension in the current pregnancy, without previous hypertension or preeclampsia were classified as having gestational hypertension.
- All those women who answered yes for preeclampsia, and no history of gestational hypertension, chronic hypertension or preeclampsia superimposed on chronic hypertension were classified as having preeclampsia.
To clarify how HDP was defined we have included a sentence on page 3, lines 96-101 (methods). The test is as follows:
Most women were enrolled before 20 weeks of gestation [28]; however, the majority of interviews were performed after 20 weeks’ gestation. Therefore, the stage of enrollment was not considered as part of the definition of HDP in our study. The criteria for classifying each HDP is detailed in Appendix S1.
A sentence was also included for presenting the proportions of women who responded the questionnaires after 20 weeks’ gestation (page 4, lines 165-166):
The majority of antenatal or perinatal questionnaires were answered after 20 weeks’ gestation (85.4%; 3641/4262).
- The proportion of women with the various forms of hypertensive disease varies and is not concordant with what is often reported in the literature. The overall frequency of HDP of 30% is rather high and is concerning for selection bias. It would benefit the manuscript to comment further on this issue. There is a 6.7% incidence of preeclampsia, which is high, but within the parameters reported in the literature. The prevalence of CHTN is 10% and for GHTN is 15% which also raises the question of validation of the diagnosis. The authors might want to comment further on this.
Response:
Although the prevalence of HDP reported in our study is higher than reported in the literature there is currently no reliable studies of the prevalence of HDP in Brazil or other LMIC. Despite the potential for misclassification of HDP, our study is the first to describe trends in the use of medications for HDP. Given the high rates of maternal mortality due to HDP in Brazil, this study is a first step in identifying how women with HDP are being managed.
The strategy adopted for identifying and including pregnancies in the Pelotas Cohort 2015 ensured that there were only 58 loss and refusals from mothers of all children born in the city of Pelotas (RS) between January 1 and December 31, 2015. Therefore, it is unlikely the prevalence of HDP as a result of selection bias. Unfortunately, this study was not designed for evaluating HDP, thus important information for ascertain the diagnosis were missed. We appreciate the considerations and comments. Therefore, we better organized the paragraph which discuss the limitations (page 11, lines 323-344). We also included data on prevalence of HDP in Brazil and validity of self-reported HDP measure.
The paragraph is presented as follows:
Our study had several limitations that should be addressed. First, there is the potential for misclassification of HDP as a result of self-reported HDP. Data on diagnoses or blood pressure measures within this population were not available and HDP were classified according to answers related to pregestational and perinatal morbidities. Logistics for collecting clinical information and recording blood pressure would probably make this type of study unfeasible. Therefore, self-reported information usually is the only feasible way for conducting large-scale studies in low-resource settings. Validation studies are thus recommended to ensure that self-reported information can be used. However, no validation studies were conducted for HDP diagnosis in this cohort. A national cross-sectional, multicenter, hospital-based study in Brazil has demonstrated moderate accuracy in self-reported HDP by pregnant women (41). The authors observed that pregnant women overreported HDP (15.9%), when compared to medical records and antenatal cards (11.1%). Second, the frequencies for overall HDP, as well as for the subtypes of HDP were higher than estimates usually reported for this type of population. Misclassification and overreporting may be a reasonable explanation. However, these may represent accurate measures of HDP in this population, considering the Pelotas Cohort enrolled almost all pregnant women in this city, which suggests that these figures cannot be explained by selections bias. We recognize the limitations on estimating any association using these data, but the proportions call for attention how pregnancies, affected or at risk of HDP, are managed. These conditions are not recorded as a basic cause of death, not reflecting the real importance and reality of its frequency (42). Mortality due to HDP is historically underreported in Brazil and Maternal Mortality Ratio due to maternal hypertensives disorders was 13 maternal deaths per 100,000 live births in 2019, suggesting that prevention strategies concerning HDP must be reevaluated (43). Finally, there is also the potential for recall bias concerning the use of medications. Although, a study by Lutz et al. (25) suggests that this is minimized by the classification of medications based on the information at the antenatal visit.
- The structure of the manuscript compares use of medication among women with and without hypertension, but the purpose of the paper is to analyze medication usage patterns among hypertensive women.
Response:
We have revised, not only the purpose, but also the title according the reviewer’s comment.
The purpose was re-written (page 2, lines 68-72):
Therefore, this study aimed to describe the patterns of use of cardiovascular agents and other medications used by women with and without HDP in the Pelotas (Brazil) Birth Cohort Study. We also aimed to describe characteristics of and to compare the patterns of use, by trimester, between both groups. A secondary objective was to describe the prevalence of maternal and perinatal outcomes in this population.
In order to ensure consistency between objective and title, a new title was proposed:
Hypertensive disorders of pregnancy and medication use in the 2015 Pelotas (Brazil) Birth Cohort Study
Response:
Figure 3B does not contribute anything to the manuscript as it reports on medications that are no antihypertensive and does not contribute to the overall understanding of the topic being discussed.
We have made revisions to the manuscript on the use of cardiovascular medications and other drugs used for managing HDP are described on methods section (page 3, lines 107-113). As we were evaluating HDP, including preeclampsia, we could not exclude the use of aspirin. Understanding if women at risk of PE are being correctly managed is of utmost importance for reducing incidence of preeclampsia.
For our study, we selected all medications whose first level of the Anatomical Therapeutic Chemical (ATC) Classification System was C (ATC - Cardiovascular System). We also selected ATC class B (Blood and Blood Forming Organs), excluding all drugs except those classified as B01AC (Platelet aggregation inhibitors excl. heparin) allowing us to evaluate acetylsalicylic acid (AAS), a medication used specifically for HDP. This approach was used in an attempt to capture treatments that could be used to manage hypertensive disorders in this population.
Considering the purpose of our study in comparing women with and without HDP, we found an interesting observation the use of aspirin and omega-3 over the pregnancy among women who report not having HDP. There is no evidence for using omega-3 for preventing HDP, an aspirin must be used by those at risk of HDP. Therefore, further studies must to be conducted to understand why these patterns were observed and in whether extension they are preventing, or not, adverse outcomes.
- Among the antihypertensive medications included in Figure 3A are isoxuprine, low dose aspirin and Omega 3 fatty acid supplements, none of which is an antihypertensive medication.
Isoxsuprine and omega-3 are classified as cardiovascular drugs and aspirin is a blood agent. The objectives were revised on page 2, lines 68-72 and in the methods on page 3, lines 107-113 to clarify the rational for including these drugs.
Specifically related to the isoxsuprine, according to ATC classification, this is a cardiovascular drug under the peripheral vasodilators group and it is used as a tocolytic agent. A systematic review has demonstrated efficacy in prolonging pregnancy in women at risk of abortion or premature delivery. Discussion about isoxsupride use was included on page 11, lines 322-326.
Isoxsuprine is a peripheral vasodilator used as a tocolytic agent, prolonging pregnancy in women at risk of abortion or premature delivery (41). In our study, women with and without HDP reported using it. The Ministry of Health guidelines recommend the use of tocolytics for premature labour, but no recommendation for using isoxsuprine was found (19), which will need further investigation.
- Clinical relevance as a descriptive study reporting on medication patterns by hypertensive women in one city of Brazil. The patterns of medication use are not necessarily generalizable to other settings were AMD is not a first line drug. If participant outcomes including modes of delivery and complications are reported, in addition to short term neonatal outcomes, this could contribute significantly to supporting use of AMD compared with other commonly used medications that lead to a more efficient response in management of HDP such as nifedipine.
The 2015 cohort is the follow-up study for all children born in the city of Pelotas (RS) between January 1 and December 31, 2015. Organized in stages, the study monitors health, physical and cognitive development and the socioeconomic context of the participants throughout life, since pregnancy. Public policies for defining breastfeeding, prevention of maternal and child mortality, nutritional status of mothers and children, physical activity, prevention of chronic diseases, among other, have used evidence provided from the Pelotas Cohorts. This population, despite being part of a cohort in Southern Brazil, are representative of the general Brazilian population and other LMIC. There is currently no information on HDP at the national level. Therefore, this work highlights the need for further research in this area.
- Lines 260-273 of the discussion are not relevant to this manuscript.
We appreciate the recommendation; however, we have revised the discussion and this paragraph highlights the lack of national guidelines updated in a regular basis for properly managing HDP. Adoption of best practices, for implementing guidelines and surveillance systems for evaluating their results are needed to reduce the burden due to HDP. If pregnancies were timely managed, adverse events could be reduced. Additionally, among all hospital admissions in this study, 30% were related to HDP, and these problems are discussed in this paragraph. The paragraph is on page 10-11, lines 251-265.
Few differences exist in terms of diagnosis and therapy management compared to high resources countries, although, like in other LMICs the problem seems to be the implementation of good-practices (16). The current guidelines for the management of pregnancies in primary care have not been updated in over 5 years (19, 30). Models for evidence-based guidelines, implementation and surveillance exist (4, 33). However, barriers to the adoption of such practices in Brazil exist.
Based on our findings, approximately 20% of pregnant women had at least one hospital admission, of which 30% were due to HDP. Although self-reported measures of admission may introduce the potential for misclassification, it is worth highlighting the potential importance and impact of this elevated rate of hospital admissions in this population and the impact on the short- and long-term health of women. Although close monitoring of pregnancies affected by or at high risk of HDP is recommended, the extent at which these women are being monitored and managed is not well understood, considering the high rates of mortality in Brazil and other LMICs. No surveillance data exist for monitoring and reducing the impact of HDP in these countries, and the burden of families affected by these conditions (34).